# The $^{15}$N gas-flux method to determine $N_2$ flux: a comparison of different tracer addition approaches

Dominika Lewicka-Szczebak[1] and Reinhard Well[2]

[1] Centre for Stable Isotope Research and Analysis, University of Göttingen, 37077 Göttingen, Germany
[2] Thünen-Institut of Climate-Smart Agriculture, Bundesallee 50, 38116 Braunschweig, Germany

*Correspondence to*: Dominika Lewicka-Szczebak (dominika.lewicka@uni-goettingen.de)

**Abstract.** The $^{15}$N gas flux method allows for the quantification of $N_2$ flux and tracing soil N transformations. An important requirement for this method is a homogeneous distribution of the $^{15}$N tracer added to soil. This is usually achieved through

soil homogenization and admixture of the $^{15}$N tracer solution or multipoint injection of tracer solution to intact soil. Both methods may create artefacts. We aimed at comparing the $N_2$ flux determined by the gas flux method using both tracer distribution approaches. Soil incubation experiments with silt loam soil using (i) intact soil cores injected with $^{15}$N label solution, (ii) homogenized soil with injected label solution and (iii) homogenized soil with admixture of label solution were performed. Intact soil cores with injected $^{15}$N tracer solution show a larger variability of the results. Homogenized soil shows

better agreement between repetitions, but significant differences in $^{15}$N enrichment measured in soil nitrate and in emitted gases were observed. For intact soil, the larger variability of measured values results rather from natural diversity of non-homogenized soil cores than from inhomogeneous label distribution. Generally, comparison of the results of intact cores and homogenized soil did not reveal statistically significant differences in $N_2$ flux determination. In both cases, a pronounced dominance of $N_2$ flux over $N_2O$ flux was noted. It can be concluded that both methods showed close agreement and

homogenized soil is not necessarily characterized by more homogenous $^{15}$N label distribution.

## 1. Introduction

Determination of soil nitrogen transformation pathways and quantification of gaseous N emissions often requires soil incubation experiments including significant manipulations of natural soil conditions. In particular, the quantification of soil $N_2$ flux in field studies is very challenging due to high atmospheric background. The most common method for both detailed tracing of soil N transformations and determination of $N_2$ emission is the application of $^{15}N$ tracer (Aulakh et al., 1991; Baily et al., 2012; Bergsma et al., 2001; Buchen et al., 2016; Deppe et al., 2017; Kulkarni et al., 2013; Morse and Bernhardt, 2013; Müller et al., 2014; Müller et al., 2004; Well et al., 2019). However, this can have a significant impact on the soil due to additional fertilization and soil disturbance depending on the method of tracer addition (Murphy et al., 2003). The impact associated with soil fertilization can be minimized by applying the lowest effective fertilizer doses. To determine soil gross N transformation rates, enrichment in $^{15}N$ of a few percent (e.g. 10 at% $^{15}N$) is sufficient (Müller et al., 2004). However, in applications where $N_2$ fluxes are analysed ($^{15}N$ gas-flux method) the labelled N pool (e.g. $NO_3^-$) should ideally be enriched by approximately 50 atom % $^{15}N$ to achieve precise results (Stevens et al., 1993). The impact of soil disturbance is often minimised by $^{15}N$ tracer application to the intact soil cores (Rütting et al., 2011).

The $^{15}N$ gas-flux method is based on the assumption of an isotopically homogenous $NO_3^-$ pool. Failure to fulfil this condition, which is often the case, may result in underestimation of denitrification rates up to 30% (Arah, 1997; Mulvaney, 1984). An initial homogeneity can be obtained through intensive mixing of the soil, but this is a massive disturbance with huge potential effects on N processes, including denitrification dynamics. However, application of intact soil cores can enhance problems with homogeneous $^{15}N$ label distribution, since incomplete equilibration of water content after injecting aqueous tracer solution could lead to increased wetness near the injection spots and to enhanced denitrification (Wu et al., 2012). Hence, for the $^{15}N$ gas-flux method a compromise must be found between homogeneous $^{15}N$ label distribution, which is crucial for $N_2$ fluxes calculations, and a possibly minimal change of the real soil N transformations.

The two most common strategies for the tracer addition to the soil are: soil homogenization where the tracer solution is mixed with the soil, or use of intact soil cores where tracer solution is added through multiple needle injections (Davidson et al., 1991). Both methods lead to potential bias. Following soil homogenization, the soil structure is changed through sieving and mixing (Gütlein et al., 2016; Kaur et al., 2010), roots and stones are removed, which should result in the best achievable homogeneity of soil properties and tracer distribution within the soil column and thus better comparability between the repetitions (Well et al., 2006). For needle injections, the soil structure stays unchanged but the pointwise injection may not ensure the homogenous distribution of the tracer (Davidson et al., 1991). Here we aimed to compare the results of these different strategies and test how far the determined $^{15}N$ pool derived $N_2$ and $N_2O$ fluxes are altered due to a particular soil treatment.

## 2. Methods

### 2.1 Experimental set-up

Silt loam soil *Albic Luvisol* from arable cropland of Merklingsen experimental station (Germany) was used (silt content
approx. 87%, 11% clay, 2% sand). Three treatments were applied: (1) soil was sieved with 4mm mesh size, the tracer
solution was added evenly, soil was homogenized and packed into the incubation column (treatment H+M: homogenized +
mixed) ; (2) intact soil cores were directly collected in the incubation columns and the tracer solution was added through the
injection needles to 12 homogeneously distributed injection points at 6 depths (in total 72 injection points per column)
(treatment I+I: intact + injected); (3) soil was sieved with 4mm mesh size (like in treatment H+M), packed into the
incubation column, and the tracer solution was added through the injection needles (like in treatment I+I) (treatment H+I:
homogenized + injected). For each treatment the soil columns were 0.3 m high with a diameter of 0.15 m. 4mm mesh size
was used because this enabled us to sieve the necessary amount of soil (56 kg) within an adequate time. The soil density of
intact cores was 1.3 g cm$^{-3}$ and the packed columns were compacted to the same density, which gave 6.89 kg soil per
column. For each soil column, 216 mL of 319 mgN L$^{-1}$ NaNO$_3$ solution with 73 at% $^{15}$N was added. This resulted in the
following initial experimental settings: 75% water-filled pores space (WFPS), 37 mg N kg$^{-1}$ NO$_3^-$, 42.5 at% $^{15}$N measured in
the subsamples of the homogenized soil immediately after tracer addition and mixing. The incubation lasted 8 days. The
columns were continuously flushed with a gas mixture with reduced N$_2$ content to increase the measurements sensitivity (2%
N$_2$ and 21% O$_2$ in He, (Lewicka-Szczebak et al., 2017)) with a flow of 10 mL min$^{-1}$. The gas samples were collected daily in
the first 4 days and every second day in the last 4 days in two 12 mL septum-capped Exetainers® (Labco Limited,
Ceredigion, UK) connected to the vents of the incubation columns.

### 2.2 Gas analyses

The gas samples were analysed with a modified GasBench II preparation system coupled with a MAT 253 isotope ratio mass
spectrometer (Thermo Scientific, Bremen, Germany) according to Lewicka-Szczebak et al. (2013). In this set-up, N$_2$O is
converted to N$_2$ prior to analysis, which allows the simultaneous measurement of stable isotope ratios $^{29}$R ($^{29}$N$_2$/$^{28}$N$_2$) and $^{30}$R
($^{30}$N$_2$/$^{28}$N$_2$), of N$_2$, of the sum of denitrification products (N$_2$+N$_2$O) and of N$_2$O. Based on these measurements the following
values were calculated according to the respective equations (after Spott et al. (2006)):

- $^{15}$N abundance of $^{15}$N-labelled pool ($a_P$), from which N$_2$ ($a_{P\_N2}$) or N$_2$O ($a_{P\_N2O}$) originate:

$$a_P = \frac{^{30}x_M - a_M \cdot a_{bgd}}{a_M - a_{bgd}} \qquad (1)$$

The calculation of $a_P$ is based on the non-random distribution of N$_2$ and N$_2$O isotopologues (Spott et al., 2006) where $^{30}x_M$ is
the fraction of $^{30}$N$_2$ in the total gas mixture:

$$^{30}x_{\mathrm{M}} = \frac{^{30}R}{1 + {}^{29}R + {}^{30}R} \tag{2}$$

$a_{\mathrm{M}}$ is $^{15}$N abundance in total gas mixture

$$a_{\mathrm{M}} = \frac{^{29}R + 2\,{}^{30}R}{2(1 + {}^{29}R + {}^{30}R)} \tag{3}$$

$a_{\mathrm{bgd}}$ is $^{15}$N abundance of non-labelled pool (atmospheric background or experimental matrix)

- the fraction originating from the $^{15}$N-labelled pool ($f_{\mathrm{P}}$) for N$_2$ ($f_{\mathrm{P\_N2}}$), N$_2$+N$_2$O ($f_{\mathrm{P\_N2+N2O}}$) and N$_2$O ($f_{\mathrm{P\_N2O}}$) within the sample:

$$f_{\mathrm{P}} = \frac{a_{\mathrm{M}} - a_{\mathrm{bgd}}}{a_{\mathrm{P}} - a_{\mathrm{bgd}}} \tag{4}$$

- N$_2$O residual fraction ($r_{\mathrm{N2O}}$) representing the unreduced N$_2$O mole fraction of pool-derived gross N$_2$O production (Lewicka-Szczebak et al., 2017).:

$$r_{\mathrm{N2O}} = \frac{y_{\mathrm{N2O}}}{y_{\mathrm{N2}} + y_{\mathrm{N2O}}} = \frac{f_{\mathrm{P\_N2+N2O}} - f_{\mathrm{P\_N2}}}{f_{\mathrm{P\_N2+N2O}}} \tag{5}$$

where $y$ represents the mole fractions.

## 2.3 Soil analyses

At the end of incubation, soil samples were collected from each column using a Goettinger boring rod with a diameter of 18 mm (Nietfeld GmbH, Quakenbrück, Germany). Three cores were taken from each column, separated into a top (0 to 15 cm) and bottom (15 to 30 cm) layer. For injected treatments ((H+I) and (M+I)) these sample cores were taken between injection points and additional cores were collected from the injection points. All soil samples were homogenised and analysed for water content (by weight loss after 24h drying in 110ºC), nitrate content (by extraction in 2M KCl 1:4) and $^{15}$N enrichment in nitrate (by bacterial denitrification method (Sigman et al., 2001)).

## 2.4 Statistics

For testing the statistical significance of the differences between treatments ANOVA and Tukey HSD Post-hoc test were applied using R 3.4.2 (R Core Team, 2013).

In Table 3, for the comparison of particular $a_{NO3}$ and $a_{\mathrm{P}}$ values, we applied the following calculated parameters:

- cumulative relative difference (cum diff) calculated as the sum of differences in $^{15}$N enrichment of different pools for all 24 samples: cum diff $= \sum_{i=1}^{n}(a_1 - a_2)_i$

- absolute mean difference (mean abs diff) calculated as the mean of modulus of differences in $^{15}$N enrichment of different pools: mean abs diff = $(\sum_{i=1}^{n} |(a_1 - a_2)_i|)/n$

In the above equations $a_1$ and $a_2$ represent the $^{15}$N enrichment of two compared pools ($a_{NO3}$ or $a_{P\_N2}$ or $a_{P\_N2O}$).

## 3. Results & Discussion

**3.1 Gas fluxes and denitrification product ratio**

In order to compare the treatments, the time course of the results must be taken into account as the gas production differed largely between the sampling dates (Fig.1). Therefore, we checked for statistically significant differences between the treatments individually for each sampling date. The results show comparable trends and no statistically significant differences between treatments (Fig.1). Notably, $r_{N2O}$ shows very good agreement at the beginning of the experiment, when

the large gas concentrations were measured, and starts to differentiate when the fluxes drop from the 3$^{rd}$ day (Fig. 1D), but these differences are not statistically significant. However, if the experiment is evaluated for the cumulative values, significant differences between treatments appear (Table 1). The cumulated gas fluxes of $N_2O$ and $N_2$ are significantly different between the treatments I+I and H+I, whereas the H+M treatment does not differ significantly from the others. However, comparison of the entire denitrification gas flux (joint $N_2$+ $N_2O$ flux) reveals no statistically significant difference

between treatments (Table 1). Product ratios are compared as cumulated $r_{N2O}$ (calculated with the cumulated fluxes) and mean $r_{N2O}$ (average value of all sampling points). Cumulated $r_{N2O}$ shows an identical pattern of significant differences as the cumulated $N_2$ and $N_2O$ fluxes. For mean $r_{N2O}$ values H+M and H+I treatment are significantly different, whereas the I+I treatment does not differ significantly from the others.

There results show that the different tracer application strategies tested had no impact on the total denitrification ($N_2$+ $N_2O$),

but the product ratio may be slightly shifted, which results in differences by comparing $N_2$ or $N_2O$ flux separately. This presumably results from the differences in distribution of moisture and nitrate between treatments (see Sect. 3.2). All determined $r_{N2O}$ values, although partially different, indicate a pronounced dominance of $N_2$ over $N_2O$ emission. Importantly, no significant differences were noted between the H+M and I+I treatment, only the H+I treatment shows higher $N_2O$ flux, lower $N_2$ flux and higher $r_{N2O}$. In this treatment we probably observe joint artefacts associated with soil homogenization and

needle injection technique.

The homogenized treatments show better comparability between the repetitions – they show lower standard deviations for gas emissions and for $r_{N2O}$ (Table 1), and smaller error bars for the daily measurements (Fig.1). The H+I treatment shows the lowest standard deviations for the cumulative gas emission measurements (Table 1). This indicates that the observed heterogeneity for I+I treatment is not due to needle injection procedure but rather due to the intact structure of soil cores,

which naturally represents the typical soil heterogeneity.

## 3.2 Soil parameters

In this study the high dose of added N resulted in more than doubled $NO_3^-$ content. This was much above the common recommendations of tracer addition of 10-25% of native soil N (Davidson et al., 1991). These recommendations are motivated by the need to minimize the fertilization effect and to trace the naturally occurring N transformation processes. But, in this study we only aimed to compare tracer addition strategies and did not intend to draw conclusions for this particular study site. Establishing a high $^{15}N$ enrichment of the $NO_3^-$ by high addition of $^{15}N$-labelled $NO_3^-$ enhanced the sensitivity of $N_2$ flux detection, which is a prerequisite for reliably identifying potential experimental artefacts, which we aimed to evaluate in this study.

A good insight into heterogeneity within columns is also provided by the soil analyses performed at the end of experiment, by collecting samples from various areas of each soil core (Table 2). Clearly, I+I treatment shows the largest standard deviations between repetitions. Also, the most pronounced differences between top and bottom soil layer can be noted for this treatment, but only soil moisture is significantly lower for the bottom layer. Since this is not the case for H+I treatment, it reflects the natural heterogeneity of intact cores rather than a result of label injection procedure. The values from injection points are never significantly different from samples between injection points (within one treatment) which indicates a good distribution of the tracer solution (Table 2).

Significant differences in soil parameters between treatments (Table 2) were observed. The I+I treatment shows significantly lower nitrate content compared to homogenized treatments (Table 2). This must be due to initial soil nitrate content. The soil was stored for two weeks before the experiment. Storing of mixed soil or sieving and homogenization procedures probably intensified N mineralization and the formation of additional nitrate through intensified nitrification, which has also been observed in previous studies (Kaur et al., 2010). Moreover, the H+M treatment shows significantly higher $^{15}N$ enrichment of $NO_3^-$ ($a^{15}N_{NO3}$) than injected treatments. This may be due to injection procedure where the needles might get partially clogged with soil causing the addition of tracer solution to be lower than planned. The assumption that the injected volume was lower than the target and thus also lower than the addition of tracer solution to H+M treatment, can also be supported by the slightly lower soil moisture and nitrate content of the injected treatments.

## 3.3 $^{15}N$ abundance in soil active pools

Despite the pronounced difference in $^{15}N$ content between treatments, the results can still be compared because the $^{15}N$ abundance of actively denitrifying pool ($a_P$ value) for each sample is individually calculated based on the distribution of $N_2$ and/or $N_2O$ isotopologues. We checked how well these calculated $a_P$ values for $N_2$ and $N_2O$ correspond with the respective $^{15}N$ enrichment measured in soil nitrate ($a_{NO3}$) and between each other (Table 3). This comparison gives additional information about the distribution of the $^{15}N$ label. The cumulative relative difference represents the overall deviation between the analyzed pools. Very high cumulative difference was noted between the $a_P$ values of both gases and $a_{NO3}$ in H+M treatment. This is mostly due to the first two sampling days, where $a_P$ values were significantly lower than $a_{NO3}$ (mean

difference of ca. 15 at% $^{15}$N, Fig.2), whereas, for the next samplings they corresponded very well (mean difference of ca. 1 at% $^{15}$N, Fig.2). This shows that initially the gases were produced in soil microsites depleted in $^{15}$N compared to the mean soil value. This is the case for all three treatments; however, the largest difference is observed for H+M treatment due to highest $a_{NO3}$ values. The absolute mean difference represents the average variation range of the compared values. For the comparison of mean absolute difference between $a_{P\_N2}$ and $a_{P\_N2O}$ we obtained quite a good agreement, much better than for the comparisons with $a_{NO3}$ (Table 3). This shows that both gases originate mostly from the same soil pool. Importantly, even in the H+M treatment where large mean difference between $a_{NO3}$ and $a_P$ values was noted, the mean difference between $a_{P\_N2}$ and $a_{P\_N2O}$ is very low. The fact that $a_{P\_N2O}$ shows much closer agreement with $a_{P\_N2}$ than $a_{NO3}$ suggests that, when missing data on $a_{P\_N2}$, which is often the case due to high $N_2$ detection limit of the gas-flux method, the $a_{P\_N2O}$ should be used rather than $a_{NO3}$ or a theoretical value on $^{15}$N abundance, as has also been proposed in previous studies (Bergsma et al., 2001; Stevens and Laughlin, 2001).

Interestingly, for the I+I treatment lower differences between $a_{NO3}$ and $a_{P\_N2O}$ or $a_{P\_N2}$ values were obtained, but larger difference between $a_{P\_N2}$ and $a_{P\_N2O}$ when compared to homogenized treatments (Table 3, Fig. 2). This shows that the multiple injection technique reduced the formation of isolated soil microsites characterized by distinct $^{15}$N enrichment when compared to the bulk $a_{NO3}$ value measured. However, the slightly higher difference between $a_P$ values for $N_2$ and $N_2O$ suggest non-identical origins for both gases, *i.e.,* probable slight admixture of hybrid $N_2$ (Spott et al., 2011) since the $^{15}$N enrichment of $N_2$ shows lower values than $N_2O$. This could explain the higher cumulated $N_2$ flux for I+I treatment (Table 1).

## 3.4 Homogeneity of $^{15}$N tracer distribution and accuracy of results

Surprisingly, the inconsistency in $^{15}$N abundance in total and actively denitrifying nitrate soil pools (Fig. 2) indicates the largest inhomogeneity at the beginning of the incubation for the homogenized soil, which is then equilibrated after 2 days of incubation. This resulted most probably from the imperfect mixing of the relatively wet (gravimetric water content of 29.3%) silt loam soil and could be due to delayed equilibration of added $^{15}$N solution into the centre of soil aggregates where denitrification rates are probably highest (Sextone et al., 1985). But, importantly, these first two days are also the ones with the highest gas production and close agreement of results between all three treatments (see Fig. 1). This suggests that even non-homogeneous distribution of $^{15}$N label and thus heterogeneity in content and $^{15}$N enrichment of nitrate in soil does not lead to severe bias in determining denitrification and its product ratio.

This study allows only for the comparison of these different treatments but not for checking the true emission values, since we have not used any independent method for fluxes determination. However, we can conclude that, despite pronounced differences in $a^{15}N$ values of different treatments and different pools, the calculated results for gas fluxes and product ratios were mostly not significantly different between the treatments. This supports the assumption that in real soil situation even imperfect label distribution allows for obtaining accurate results (Arah, 1997; Davidson et al., 1991; Deppe et al., 2017). But, importantly, this is possible only if we measure and use $a_P$ values representing the $^{15}$N values of the pools actively producing $N_2$ and $N_2O$. The fluxes would be significantly underestimated if the $a_{NO3}$ value was applied for calculations, *e.g.*, for the first

sampling point this would result in about 20% underestimation of the $N_2$ flux when the measured final $a_{NO3}$ value was applied, and about 30% underestimation when the initial $a_{NO3}$ value was applied. Significant differences in $^{15}N$ enrichment of total and active nitrate pool has also been found in our previous laboratory and field studies (Buchen et al., 2016; Deppe et al., 2017). It was shown that in such cases the $^{15}N$ enrichment of N pool undergoing denitrification is well represented by $a_P$

values, but not by $a_{NO3}$ values.

The homogeneity of $^{15}N$ label distribution depends not only on the tracer addition technique but even more on the soil type, water content, initial nitrate and ammonium content. In our previous laboratory experiments quite a good agreement between $a_{NO3}$ values and $a_P$ values was achieved indicating a homogenous denitrifying pool (Lewicka-Szczebak et al., 2017). In that study similar soil texture was used (silt loam), but the initial amount of nitrate and ammonium was very low, and soil

samples were prepared at soil moisture of 70% WFPS with rest water added on top, and soil was incubated in high moisture conditions. But notably, the anoxic conditions showed perfect agreement in $a_{NO3}$ and $a_P$ values whereas for oxic conditions slight differences have also been noted (Lewicka-Szczebak et al., 2017). Oxic conditions can be expected to yield greater disagreement between $a_{NO3}$ and $a_P$ due to dilution of the bulk $a_{NO3}$ by soil-derived nonlabelled N sources in contrast to anoxic soil microsites (Deppe et al., 2017). In the H+M treatment of the actual experiment, inhomogeneity was probably the result

of soil moisture during soil homogenization being too high (75% WFPS) causing the formation of larger aggregates. But this problem can be overcome if the $^{15}N$ label is incorporated at low soil moisture and target moisture is established by adding water afterwards (Lewicka-Szczebak et al., 2017, Well et al., 2019).

**Conclusions**

Soil homogenisation reduced the variability within the soil column and between repetitions but not necessarily improved the

$^{15}N$ label distribution. Wet homogenisation has led to uneven label and process distribution. Multiple needle injections of $^{15}N$ solution resulted in better agreement between $^{15}N$ enrichment of soil and emitted gases, indicating even more homogeneous $^{15}N$ label distribution than homogenised treatments.

Larger heterogeneity of intact soil cores, noted as larger deviations of all measured values, reflects the natural soil conditions rather than inhomogeneous $^{15}N$ label distribution. Importantly, the results obtained with homogenised soil and with intact

soil cores do not differ significantly in the determined $N_2$ flux and denitrification product ratio. Hence, when applying each of these treatments, very similar general conclusions will be found, *i.e.,* the dominance of the $N_2$ flux over the $N_2O$ flux. This similarity in the results is thanks to the calculation method applying $a_P$ values determined individually for each sample which assures the adequate results for flux calculation, even with the existence of multiple N pools. It was found that $a_{NO3}$ values can differ greatly from the $a_P$ value of produced gases and its application for $N_2$ flux determination may result in large bias.

In this study only one soil with one moisture level was tested and this experiment was conducted with high doses of $^{15}N$ labeled fertilizer. Since the indicated artefacts due to homogenisation and mixing depend on soil properties such as organic matter properties, pore structure, microbial community dynamics or heterogeneity of label and water distribution, for more

universal conclusions further studies with different soils, moistures and [15]N label additions should be conducted. Meanwhile, to minimize methodical bias in future studies using the [15]N gas flux method, our approach could be used to test labelling

artefacts for specific soil conditions.

**Data availability.** Original data is available upon request. Material necessary for this study's findings is presented in the paper.

**Author contribution**. DLS and RW designed the experiment and DLS carried it out. Both authors interpreted the results. DLS prepared the manuscript with significant contribution from RW.

*Competing interests.* The authors declare that they have no conflict of interest.

*Acknowledgements.* This study was financed by German Research Foundation (DFG: LE 3367/1-1). Many thanks are due to Frank Hegewald and Nicolas Ruoss for help in the collection of soil cores and setting up laboratory incubations, Stefan Burkart for help in carrying out soil incubation, Martina Heuer for help in isotopic analyses, Nicole Altwein and Ute Tambor for help in soil analyses, Kerstin Gilke for help in chromatographic analyses.

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

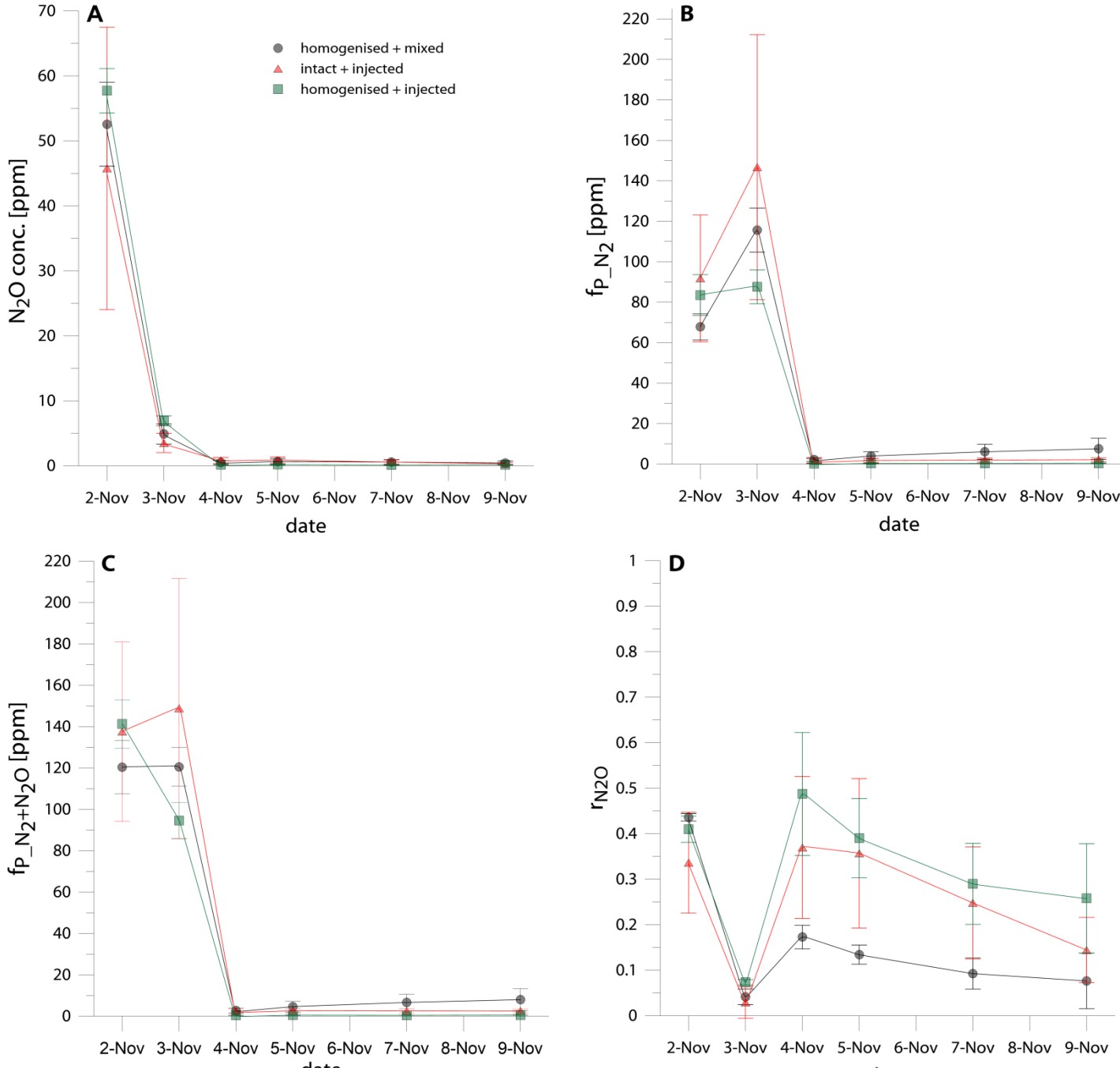

**Figure 1: Comparison of the temporal changes in N₂O concentration (A), fraction of ¹⁵N-pool derived N₂ (B), fraction of ¹⁵N-pool derived denitrification products (N₂+N₂O) (C), and N₂O residual fraction (D) in three treatments: homogenized soil mixed with fertilizer (black dots), intact soil cores with fertilizer added through needle injection (red triangles), and homogenized soil with fertilizer added through needle injection (green squares). Error bars represent the standard deviation of 4 replicates within one treatment.**

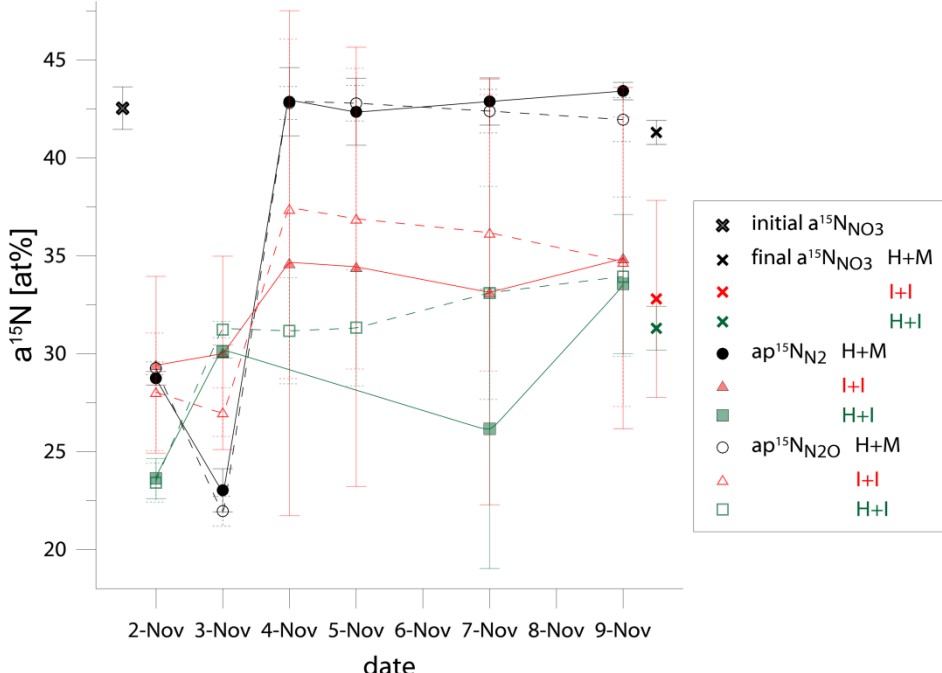

**Figure 2: Comparison of $^{15}N$ abundance in total initial and final soil nitrate ($a^{15}N_{NO3}$) and in active soil pool emitting $N_2$ ($a_P^{15}N_{N2}$) and $N_2O$ ($a_P^{15}N_{N2O}$) in three treatments: homogenized soil and mixed fertilizer (H+M, black points)), intact soil core and injected fertilizer (I+I, red points), homogenized soil and injected fertilizer (H+I, green points).**


**Table 1: Comparison of cumulated fluxes, cumulated product ratio (cum $r_{N2O}$) and mean product ratios (mean $r_{N2O}$) in three treatments: homogenized and mixed (H+M), intact and injected (I+I), homogenized and injected (H+I). Statistically significant differences are indicated (\*p<0.05, \*\*p<0.01, \*\*\*p<0.001).**

| treatment | cum $N_2O$ [mgN kg soil$^{-1}$ day$^{-1}$] | | cum $N_2$ [mgN kg soil$^{-1}$ day$^{-1}$] | | cum $N_2$+$N_2O$ [mgN kg soil$^{-1}$ day$^{-1}$] | | cum $r_{N2O}$ | | mean $r_{N2O}$ | |
|---|---|---|---|---|---|---|---|---|---|---|
| **H+M** | $0.63 \pm 0.10$ | ab | $2.16 \pm 0.31$ | ab | $2.80 \pm 0.38$ | a | $0.23 \pm 0.05$ | ab | $0.16 \pm 0.14$ | a |
| **I+I** | $0.55 \pm 0.26$ | a | $2.62 \pm 1.08$ | a | $3.16 \pm 1.18$ | a | $0.18 \pm 0.14$ | a | $0.25 \pm 0.14$ | ab |
| **H+I** | $0.69 \pm 0.05$ | b\*\* | $1.83 \pm 0.20$ | b\* | $2.53 \pm 0.23$ | a | $0.27 \pm 0.04$ | b\*\* | $0.32 \pm 0.15$ | b\*\*\* |


**Table 2: Soil analyses at the end of the experiment: mixed samples, and separately from the top and bottom layer and for injected columns also from injection points (including both top and bottom layer). Statistically significant differences are indicated with uppercase letters (\*\*p<0.01, \*\*\*p<0.001). For individual values the differences within treatment were tested, and for mean values the differences between treatments were tested.**

| treatment | sample | WFPS [%] | mean WFPS [%] | $NO_3^-$ conc. [mg N kg$^{-1}$] | mean $NO_3^-$ conc. [mg N kg$^{-1}$] | $a^{15}N_{NO3}$ [at%] | mean $a^{15}N_{NO3}$ [at%] |
|---|---|---|---|---|---|---|---|
| **H+M** | top | $71.5 \pm 0.4$ [a] | $71.8 \pm 0.6$ [a] | $35.5 \pm 0.5$ [a] | $35.4 \pm 0.4$ [a] | $41.2 \pm 0.5$ [a] | $41.3 \pm 0.4$ [a]\*\*\* |
| | bottom | $72.1 \pm 0.8$ [a] | | $35.2 \pm 0.3$ [a] | | $41.3 \pm 0.3$ [a] | |
| **I+I** | top | $72.3 \pm 2.0$ [a]\*\* | $68.9 \pm 3.5$ [a] | $28.6 \pm 5.5$ [a] | $25.8 \pm 4.6$ [b]\*\*\* | $29.3 \pm 2.7$ [a] | $32.8 \pm 5.9$ [b] |
| | bottom | $65.3 \pm 1.8$ [b]\*\* | | $22.5 \pm 2.7$ [a] | | $36.1 \pm 7.0$ [a] | |
| | injection point | $69.0 \pm 1.9$ [ab] | | $26.4 \pm 3.6$ [a] | | $33.0 \pm 6.5$ [a] | |
| **H+I** | top | $69.7 \pm 2.3$ [a] | $69.6 \pm 1.9$ [a] | $32.6 \pm 0.4$ [a] | $32.1 \pm 1.5$ [a] | $30.9 \pm 1.2$ [a] | $31.3 \pm 3.0$ [b] |
| | bottom | $70.2 \pm 1.3$ [a] | | $33.0 \pm 0.8$ [a] | | $33.7 \pm 1.8$ [a] | |
| | injection point | $68.8 \pm 2.1$ [a] | | $30.7 \pm 1.7$ [a] | | $29.2 \pm 3.9$ [a] | |


**Table 3: Differences between the measured $^{15}N$ abundance in soil nitrate ($a_{NO3}$) and determined $^{15}N$ abundance of $^{15}N$-pool derived $N_2$ ($a_{P\_N2}$) and $N_2O$ ($a_{P\_N2O}$) expressed as the cumulative relative difference for all samples (n=24), mean absolute difference (see section 2.4 for calculation procedure). In the above equations $a_1$ and $a_2$ represent the $^{15}N$ enrichment of two compared pools ($a_{NO3}$ or $a_{P\_N2}$ or $a_{P\_N2O}$).**


| difference | $a_{NO3}$- $a_{P\_N2}$ | | $a_{NO3}$- $a_{P\_N2O}$ | | $a_{P\_N2O}$- $a_{P\_N2}$ | |
|---|---|---|---|---|---|---|
| | cum diff [$^{15}N$ at%] | mean abs diff [$^{15}N$ at%] | cum diff [$^{15}N$ at%] | mean abs diff [$^{15}N$ at%] | cum diff [$^{15}N$ at%] | mean abs diff [$^{15}N$ at%] |
| **H+M** | 99 | 7.8 | 107 | 6.1 | -8 | 2.3 |
| **I+I** | 1 | 6.3 | -14 | 5.3 | 15 | 3.4 |
| **H+I** | 53 | 4.2 | 18 | 3.0 | 37 | 2.4 |

