# Peer review of "The 15N gas-flux method to determine $N_2$ flux: a comparison of different tracer addition approaches"

_SOIL, 2019_

## Referee Comment (RC1) · Anonymous Referee #1 · 7 Jan 2020

This is an informative and relevant study, the experiments are well planned and conclusions are sound. Prior to publication, a few clarifications are needed. The paper would also benefit from language editing (e.g. past and present tense are mixed).

General comments: Both, the introduction and discussion could benefit from including references that support your statements. There are quite a few statements, which are unsupported by references and/or your results. Although this might be the first paper on the effect of 15N tracer approach on the N gas source partitioning, some other papers have investigated the effects of tracer addition on the soil N cycle (Davidson et al.,

1991; Gütlein et al., 2016; Kaur et al., 2010). It might be worth looking at those (you do not need to cite those necessarily, but they might contribute to your discussion). The tracer addition (with a 15N fraction of 73 %), resulted in an initial 15N fraction of soil NO3- of 42.5 % (line 51). This means that soil NO3- content was more than doubled, which is much above common recommendations of tracer addition (10 – 25 % of native soil N). What was the motivation for such a high addition of tracer and what are the consequences for your results? I would like to see a discussion on this. Your comparison of the 15N fraction of NO3- (a_NO3) with the calculated a_p values (line 127) makes only sense if NO3- was the sole source of N2O and n2, i.e. all gases were produced via denitrification. What supports this assumption? You speculate yourself later about the possibility for hybrid N2 (line 148). And N2O production from nitrification is also possible.

Specific comments: Abbreviations should be introduced at first use and then used consequently Do not start sentences with abbreviation or chemical symbols (e.g. line 8) Line 11: please be more specific what kind of results. Line 13: "wider range" is unclear, be more specific. Line 40: I suggest you first describe the soil, before describing the treatments, i.e. start this paragraph with text from line 47. Line 49: this is an unusual unit for soil density; maybe use the more common cm-3? Line 51: what is "initial condition"? Is this prior to trace addition or immediately after? Please clarify. Line 66: the ap values, are those calculated or measured? I think this part would benefit from showing all equations rather than referring solely to other papers. Line 102: This sentence needs rephrasing; "we may deal with" is unclear. Line 110 (&114): The phrase "column heterogeneity" is unclear and might be confusing. As I understand you mean the heterogeneity between different columns, but it sounds like the within column heterogeneity. The latter, you actually cannot conclude about. Line 115: Suggest adding "(Table 2)" at the end of sentence. Line 116: "Very" is imprecise. Avoid such qualitative statements. Line 117: For me it is unclear why the initial NO3- content should differ between the treatments. After all, it is the same soil. Alternatively, it might be due to stimulated nitrification in the mixed soil (see e.g. Kaur et al., 2010). Line

119-123: This sounds somewhat unlikely to me. If less 15N was injected, you certainly should have noted that during the injections. Line 129 (& 136): Suggest moving the text in parentheses (after colon) to the Methods. Line 131 & 144: The "differences" you refer to, is this the cumulative or mean? Line 171: should rather be "tracer addition" Line 172: here you use for the first time "content" of inorganic N, while otherwise you use concentration. In fact, content is the correct term.

Figure 1, caption: "black dots" Figure 1, caption: Last sentence not needed (as there are no statistical differences) Table 2: Unclear what is compared statistically, within treatment of between? Also, what is the "mean" referring to, mean of what? The "Injection point", is this for both layers? Table 3: Suggest moving the equations (with additional explanations) to the method section.

Cited references: Davidson, E.A., Hart, S.C., Shanks, C.A., Firestone, M.K., 1991. Measuring gross nitrogen mineralization, immobilization, and nitrification by 15N isotopic pool dilution in intact soil cores. Journal of Soil Science 42, 335-349. Gütlein, A., Dannenmann, M., Kiese, R., 2016. Gross nitrogen turnover rates of a tropical lower montane forest soil: Impacts of sample preparation and storage. Soil Biology and Biochemistry 95, 8-10. Kaur, A.J., Ross, D.S., Fredriksen, G., 2010. Effect of soil mixing on nitrification rates in soils of two deciduous forests of Vermont, USA. Plant and Soil 331, 289-298.
* * *

---

## Referee Comment (RC2) · Anonymous Referee #2 · 30 Jan 2020

General comments:

This is a short communication on a comparison study of the effect of two different 15N tracer application techniques, i.e. mixing of tracer with soil and injection of tracer into the soil, on N2O and N2 fluxes. They used either undisturbed soil cores or disturbed, sieved soil, recompacted back to the original bulk density after homogenization. The authors measured N2O and N2 evolution from the soil after 15N tracer (nitrate) application on six different days over a period of eight days. They found generally no significant differences in N2 flux between intact soil cores and homogenized soil, with

strong dominance of N2 over N2O fluxes. The larger variability of N gas fluxes found in intact soil cores was attributed to the natural heterogeneity of soil. The paper is very short, which is not a minus in itself, as it is on an interesting and relevant topic. The idea to compare 15N label injection to intact or homogenized soil with prior mixing of the label with homogenized soil is original. Nevertheless, the paper appears to be at a premature stage, as only one soil type was studied at one water level (75% WFPS), and as the 15N label was applied at a relatively high dose (more than 100% of the natural soil nitrate pool, as indicated by the initial 15N content of the nitrate pool immediately after addition of the label), which might have strongly biased the obtained results. Therefore, I suggest that the authors conduct additional experiments with different soils, at different water levels, and with lower doses of 15N label, and evaluate the results on this broader basis of results

Specific comments:

Title: The title suggests that N2O pathways have been characterized in the study, implying that also N2O production pathways, e.g. either from nitrification or from denitrification have been elucidated, which was not really the case.

Abstract: It does not become clear from the Abstract, whether this is a (mini-)review or whether only own results were compared. Furthermore, the Abstract does not provide any information about the experimental setup. In L16-19 it should be indicated for which soil the results were obtained.

Introduction: The introduction is very short. Despite the statement in L27-28 that the 15N tracer application technique "implies a significant impact for the soil due to additional fertilization and soil disturbance depending on the way of tracer addition", and the fact that exactly this technique was applied in the present study, no further elaboration of this topic follows. Thus, some further information from the literature should be added here.

Materials and Methods: L41: no rationale has been provided why the soil was sieved

at 4 mm, and not e.g. at 2 mm, as commonly done.

L61: The ratio 30R should be 30N2/28N2, not 30N2/29N2

L116: Not clear which differences in what were observed here.

L 136: "modulus of differences": Isn't the modulus the rest of a division?

L137: "Here it clear...": Unclear at this point, what is clear why.

L 138-139: "...much better than for comparisons with aNO3 (Table 3). This shows that both gases originate mostly from the same soil pool.": But the pool they originate from is the nitrate pool, isn't it? Shouldn't all three parameter be then comparable with each other?

L146: "...than the aNO3 value measured for total soil.": The logic of this part of the sentence is not clear.

L160-162: Check wording, this sentence is hard to understand.

L173-175: I would have expected the opposite logic here, i.e. that oxic conditions lead to greater disagreement due PRESENCE of nitrification and hence MORE dilution of the 15N-nitrate pool by native (soil-derived) N-sources.

L191-193: I think also here the logic is wrong. As it stands, the dominance of N2 fluxes is due to the calculation method applied.

Figures general: I would not recommend the use of spline functions to connect the data points, but the use of straight lines instead.

Fig. 1: Caption and figure panels do not fit together. Caption 1B says "fraction of 15N-pool derived N2O", but Fig. 1B shows fp_N2, but the values are in ppm, which does not make sense (should be dimensionless between 0 and 1). Caption 1C says "N2 concentration", but Fig. 1C shows fp_N2+N2O, and again the values are in ppm, but should be dimensionless between 0 and 1.

Technical corrections: Can be found in the annotated pdf submitted with this review.

Please also note the supplement to this comment:
https://www.soil-discuss.net/soil-2019-64/soil-2019-64-RC2-supplement.pdf

───────────────────────────────

[Figure]

**Supplement:**

[revised manuscript text omitted]

---

## Author Comment (AC1) · 6 Feb 2020

*Review response for Anonymous referee #1 on*

**$^{15}$N gas-flux method to determine $N_2$ emission and $N_2O$ pathways: a comparison of different tracer addition approaches**

by Dominika Lewicka-Szczebak and Reinhard Well

[5]

- [(1)] *comments from referees*
- [(2)] authors response
- [(3)] authors changes in manuscript

[10]

*This is an informative and relevant study, the experiments are well planned and conclusions are sound. Prior to publication, a few clarifications are needed. The paper Gould also benefit from language editing (e.g. past and present tense are mixed).*

Thank you. We will make the clarifications needed and the professional language editing will be performed after
[15] the needed corrections and their acceptance.

*Both, the introduction and discussion could benefit from including references that support your statements. There are quite a few statements, which are unsupported by references and/or your results. Although this might be the first paper on the effect of 15N tracer approach on the N gas source partitioning, some other papers have*
[20] *investigated the effects of tracer addition on the soil N cycle (Davidson et al., 1991; Gütlein et al., 2016; Kaur et al., 2010). It might be worth looking at those (you do not need to cite those necessarily, but they might contribute to your discussion).*

Thank you for the very adequate citation suggestions. These and further references will be included in the manuscript introduction and discussion:

[25] Introduction: line 32 (Davidson et al., 1991), line 33(Gütlein et al., 2016; Kaur et al., 2010), line 36 (Davidson et al., 1991),
Discussion: line 163 (Davidson et al., 1991), line 118: (Kaur et al., 2010).

*The tracer addition (with a 15N fraction of 73 %), resulted in an initial 15N fraction of soil NO3- of 42.5 % (line*
[30] *51). This means that soil NO3- content was more than doubled, which is much above common recommendations of tracer addition (10 – 25 % of native soil N). What was the motivation for such a high addition of tracer and what are the consequences for your results? I would like to see a discussion on this.*

The reason for high N addition was the limited sensitivity of 15N gas flux method. The N2 gas flux is only detectable for the high 15N content. The common recommendations for low N additions are important for the
[35] studies where we want to trace the natural N transformation for this soil and the fertilization effect must be as

minimal as possible. Here our aim was to compare the effects of the method of tracer addition, i.e. homogenisation vs injection, so it was important to obtain a well detectable N2 flux and it was not intended to draw conclusions on the denitrification activity for the particular study site. If we compare the different addition strategies by addition of even more N than usual, the potential experimental artefacts should be even enhanced, which would be a positive consequence for our study objectives. This discussion will be added to the manuscript at the beginning of 3.2 section, line 110 :

In this study the addition of N to the soil was quite high resulting in more than doubled NO3- content. This was much above the common recommendations of tracer addition of 10-25% of native soil N (Davidson et al., 1991). These recommendations are motivated by the need of minimizing the fertilization effect to trace the naturally occurring N transformation processes. But, in this study we only aimed at comparison of tracer addition strategies and not intended to draw conclusions for this particular study site. Establishing a high $^{15}$N enrichment of the $NO_3^-$ by high addition of $^{15}$N-labelled $NO_3^-$ enhanced the sensitivity of $N_2$ fluxes detection, which is a prerequisite for reliably identifying potential experimental artefacts, which we aimed to evaluated in this study.

We will also add this information in the introduction:

To determine soil gross N transformation rates, enrichment in $^{15}$N of a few percent (e.g. 10 at% $^{15}$N) is sufficient (Müller et al., 2004). However, in applications where $N_2$ fluxes are analysed ($^{15}$N gas-flux method) the labelled N pool (e.g. $NO_3^-$) should ideally be enriched by approximately 50 at% $^{15}$N to achieve precise results (Stevens et al., 1993).

*Your comparison of the 15N fraction of NO3- (a_NO3) with the calculated a_p values (line 127) makes only sense if NO3- was the sole source of N2O and n2, i.e. all gases were produced via denitrification. What supports this assumption? You speculate yourself later about the possibility for hybrid N2 (line 148). And N2O production from nitrification is also possible.*

Quite a high soil moisture favours denitrification. We only labelled the nitrate pool so when calculating aP this refers to labelled pool, nitrate. Other gas sources, originating from unlabelled pools, like eg. nitrification, are obtained from the isotope ratios of emitted N2O (data not shown).
If hybrid gases are present the aP values are lower than nitrate a15N. That's why we speculate either about heterogenity or hybrid gas production.

*Specific comments*

All the specific comments have been taken into account and the relevant changes will be incorporated into the manuscript

*Line 11: please be more specific what kind of results.*

We aimed at comparing the N2 flux determined by the gas flux method

*Line 13: "wider range" is unclear, be more specific.*

It will be changed to: larger variability

*Line 51: what is "initial condition"? Is this prior to trace addition or immediately after? Please clarify.*

This will be clarified:
measured in the subsamples of the homogenized soil immediately after tracer addition and mixing

*Line 66: the ap values, are those calculated or measured? I think this part would benefit from showing all equations rather than referring solely to other papers.*

The equations will be added:

Based on these measurements the following values are calculated according to the respective equations (after Spott et al. (2006)):

- $^{15}N$ abundance of $^{15}N$-labelled pool ($a_P$) from which $N_2$ ($a_{P\_N2}$) or $N_2O$ ($a_{P\_N2O}$) originate:

$$a_P = \frac{^{30}x_M - a_M \cdot a_{bgd}}{a_M - a_{bgd}} \tag{1}$$

The calculation of $a_P$ is based on the non-random distribution of $N_2$ and $N_2O$ isotopologues (Spott et al., 2006) where $^{30}x_M$ is the fraction of $^{30}N_2$ in the total gas mixture:

$$^{30}x_M = \frac{^{30}R}{1 + {^{29}R} + {^{30}R}} \tag{2}$$

$a_M$ is $^{15}N$ abundance in total gas mixture

$$a_M = \frac{^{29}R + 2\ {^{30}R}}{2(1 + {^{29}R} + {^{30}R})} \tag{3}$$

$a_{bgd}$ is $^{15}N$ abundance of non-labelled pool (atmospheric background or experimental matrix)

- the fraction originating from the $^{15}N$-labelled pool ($f_P$) for $N_2$ ($f_{P\_N2}$), $N_2 + N_2O$ ($f_{P\_N2+N2O}$) and $N_2O$ ($f_{P\_N2O}$) within the sample:

- $$f_P = \frac{a_M - a_{bgd}}{a_P - a_{bgd}} \tag{4}$$

- $N_2O$ residual fraction ($r_{N2O}$) representing the unreduced $N_2O$ mole fraction of pool-derived gross $N_2O$ production (Lewicka-Szczebak et al., 2017).:

- $$r_{N2O} = \frac{y_{N2O}}{y_{N2} + y_{N2O}} = \frac{f_{P\_N2+N2O} - f_{P\_N2}}{f_{P\_N2+N2O}} \tag{5}$$

110      -    where *y* represents the mole fractions.

*Line 102: This sentence needs rephrasing; "we may deal with" is unclear.*

This will be rephrased to: we probably observe

115

*Line 110 (&114): The phrase "column heterogeneity" is unclear and might be confusing. As I understand you mean the heterogeneity between different columns, but it sounds like the within column heterogeneity. The latter, you actually cannot conclude about.*

120 This is heterogenity within one column, determined at the end of experiment by destructive sampling of multiple samples within one column. This will be clarified, by using 'heterogeneity within columns'

*Line 117: For me it is unclear why the initial NO3- content should differ between the treatments. After all, it is the same soil. Alternatively, it might be due to stimulated nitrification in the mixed soil (see e.g. Kaur et al.,*
125 *2010).*

This is due to storage, sieving and homogenisation - same as indicated by Kaur et al, 2010. Thank you for information on this paper! Explanation and citation will be added:

130 Storing of mixed soil or sieving and homogenization procedures probably intensified N mineralization and formation of additional nitrate through intensified nitrification, which has been also observed in previous studies (Kaur et al., 2010).

*Line 119-123: This sounds somewhat unlikely to me. If less 15N was injected, you certainly should have noted*
135 *that during the injections.*

This could have not been noted during the injections. For all columns 3L of solution were prepared, this included 400mL reserve above the calculated needed amount (needed e.g. for flushing the needles before injection). I didn't measured exactly the amount lost during injection and left after injection, hence I also wasn't able to assess
140 the unplanned losses during the injection.

*Line 129 (& 136): Suggest moving the text in parentheses (after colon) to the Methods.*

This will be moved to the methods section 2.4
145

In Table 3 for the comparison of particular $a_{NO3}$ and $a_P$ values we applied following calculated parameters:

     -    cumulative relative difference (cum diff) calculated as a sum of differences in [15]N enrichment of different pools for all 24 samples: cum diff = $\sum_{i=1}^{n}(a_1 - a_2)_i$

     -    absolute mean difference (mean abs diff) calculated as a mean of modulus of differences in [15]N
150          enrichment of different pools: mean abs diff = $(\sum_{i=1}^{n}|(a_1 - a_2)_i|)/n$

In the above equations $a_1$ and $a_2$ represent the [15]N enrichment of two compared pools ($a_{NO3}$ or $a_{P\_N2}$ or $a_{P\_N2O}$).

*Line 131 & 144: The "differences" you refer to, is this the cumulative or mean?*

155   First cumulative and later mean. This will be added in the text.

*Line 172: here you use for the first time "content" of inorganic N, while otherwise you use concentration. In fact, content is the correct term.*

160   This will be corrected for content in the whole manuscript.

*Table 2: Unclear what is compared statistically, withintreatment of between? Also, what is the "mean" referring to, mean of what? The "Injection point", is this for both layers?*

165   This caption will be modified

Table 2: Soil analyses at the end of the experiment: mixed samples, and separately from the top and bottom layer and for injected columns also from injection points (including both top and bottom layer). Statistically significant differences are indicated with uppercase letters (**p<0.01, ***p<0.001). For individual values, the differences
170   within treatment and for mean values the differences between treatments were tested.

*Table 3: Suggest moving the equations (with additional explanations) to the method section.*

They will be moved to the section 2.4.

175

---

## Author Comment (AC2) · 6 Feb 2020

*Review response for Anonymous referee #2 on*

**$^{15}$N gas-flux method to determine N$_2$ emission and N$_2$O pathways: a comparison of different tracer addition approaches**

by Dominika Lewicka-Szczebak and Reinhard Well

- [1] *comments from referees*
- [2] authors response
- [3] authors changes in manuscript

*This is a short communication on a comparison study of the effect of two different 15N tracer application techniques, i.e. mixing of tracer with soil and injection of tracer into the soil, on N2O and N2 fluxes. They used either undisturbed soil cores or disturbed, sieved soil, recompacted back to the original bulk density after homogenization. The authors measured N2O and N2 evolution from the soil after 15N tracer (nitrate) application on six different days over a period of eight days. They found generally no significant differences in N2 flux between intact soil cores and homogenized soil, with strong dominance of N2 over N2O fluxes. The larger variability of N gas fluxes found in intact soil cores was attributed to the natural heterogeneity of soil. The paper is very short, which is not a minus in itself, as it is on an interesting and relevant topic. The idea to compare 15N label injection to intact or homogenized soil with prior mixing of the label with homogenized soil is original. Nevertheless, the paper appears to be at a premature stage, as only one soil type was studied at one water level (75% WFPS), and as the 15N label was applied at a relatively high dose (more than 100% of the natural soil nitrate pool, as indicated by the initial 15N content of the nitrate pool immediately after addition of the label), which might have strongly biased the obtained results. Therefore, I suggest that the authors conduct additional experiments with different soils, at different water levels, and with lower doses of 15N label, and evaluate the results on this broader basis of results.*

Thank you for the positive comments. We fully agree that testing the relevance of labelling techniques on measured denitrification should be extended to other conditions and soil types since the suspected artefacts by homogenisation and mixing depend on soil properties such as organic matter properties, pore structure, microbial community dynamics or heterogeneity of label and water distribution. This test was performed for the only one soil type that we needed for our further studies of a certain project to evaluate the comparability of the results and answer the question if the injection technique may cause bias of the results. High soil moisture and high enrichment of nitrate was necessary to enhance denitrification and optimizes measuring sensitivity in view of the poor sensitivity of the 15N gas flux method. Please note that in past denitrification studies using the 15N gas flux method, these potential artefacts have been ignored. Therefore we think it is useful to publish these first results. A study large enough in terms of soil types and conditions which would allow to generalise our findings representing all possible conditions would be far beyond the feasibility of our current project. It would be certainly very interesting, but currently we do not have resources for performing this. Therefore, we believe this short study is worth publishing as the first idea which should be deepened by future studies. This need for further research will be emphasised at the end of conclusions:

In this study only one soil with one moisture level was tested and this experiment was conducted with high dose of $^{15}$N labeled fertilizer. Since the indicated artefacts due to homogenisation and mixing depend on soil properties such as organic matter properties, pore structure, microbial community dynamics or heterogeneity of label and water distribution, for more universal conclusions further studies with different soils, moistures and $^{15}$N label additions should be conducted.

Due to the exemplary character of our study we submitted it as a short communication.

The high dose of 15N label applications was applied due to the limited sensitivity of 15N gas flux method. The N2 gas flux is only detectable for the high 15N content. The common recommendations for low N additions are important for the studies where we want to trace the natural N transformation for this soil and the fertilization effect must be as minimal as possible. Here our aim was to compare the effects of tracer addition, so it was important to obtain a well detectable N2 flux and it was not intended to draw conclusions for the particular study site. If we compare the different addition strategies by addition of even more N than usual, the potential experimental artefacts should be even enhanced, which would be positive consequence for our study objectives. This discussion will be added to the manuscript at the beginning of 3.2 section, line 110 :

In this study, the addition of N to the soil was quite high resulting in more than doubled NO3- content. This was much above the common recommendations of tracer addition of 10-25% of native soil N (Davidson et al., 1991). These recommendations are motivated by the need of minimizing the fertilization effect to trace the naturally occurring N transformation processes. But, in this study we only aimed at comparison of tracer addition strategies and not intended to draw conclusions for this particular study site. Establishing a high 15N enrichment of the NO3- by high addition of 15N-labelled NO3- enhanced the sensitivity of N2 fluxes detection, which is a prerequisite for reliably identifying potential experimental artefacts, which we aimed to evaluated in this study.

We will also add this information in the introduction:

To determine soil gross N transformation rates, enrichment in $^{15}$N of a few percent (e.g. 10 at% $^{15}$N) is sufficient (Müller et al., 2004). However, in applications where $N_2$ fluxes are analysed ($^{15}$N gas-flux method) the labelled N pool (e.g. $NO_3^-$) should ideally be enriched by approximately 50 at% $^{15}$N to achieve precise results (Stevens et al., 1993).

*Specific comments:*

*Title: The title suggests that N2O pathways have been characterized in the study, implying that also N2O production pathways, e.g. either from nitrification or from denitrification have been elucidated, which was not really the case.*

The title will be changed accordingly:

The $^{15}$N gas-flux method to determine $N_2$ flux : a comparison of different tracer addition approaches

*Abstract: It does not become clear from the Abstract, whether this is a (mini-)review or whether only own results were compared. Furthermore, the Abstract does not provide any information about the experimental setup. In L16-19 it should be indicated for which soil the results were obtained.*

The missing information in the abstract will be added, line 13:

Soil incubation experiments with silt loam soil using (i) intact soil cores injected with 15N label solution and (ii) homogenized soil with injected label solution and (iii) homogenized soil with admixture of label solution were performed.

*Introduction: The introduction is very short. Despite the statement in L27-28 that the 15N tracer application technique "implies a significant impact for the soil due to additional fertilization and soil*

*disturbance depending on the way of tracer addition", and the fact that exactly this technique was applied in the present study, no further elaboration of this topic follows. Thus, some further information from the literature should be added here.*

Introduction will be expanded by adding (after line 28):

The impact associated with soil fertilization can be minimized by applying the lowest effective fertilizer doses. In most cases, enrichment in $^{15}$N of a few percent (e.g. 10 at% $^{15}$N) is sufficient to determine soil N transformation rates (Müller et al., 2004). However, in applications where gaseous N species such as $N_2O$ and $N_2$ are analysed ($^{15}$N gas-flux method) the labelled N pool (e.g. $NO_3^-$) should ideally be enriched by approximately 50 at% $^{15}$N which provides the most precise results (Stevens et al., 1993). The impact due to soil disturbance is often minimised by $^{15}$N tracer application to the intact soil cores (Rütting et al., 2011).

The $^{15}$N gas-flux method is based on the assumption of an isotopically homogenous $NO_3^-$ pool. Failure to fulfil this condition, which is often the case, may result in underestimation of denitrification rates up to 30% (Arah, 1997; Mulvaney, 1984). An initial homogeneity can be obtained by intensive mixing of the soil, but this is a massive disturbance with huge potential effects on N processes including denitrification dynamics. However, application of intact soil cores can enhance problems with homogeneous $^{15}$N label distribution, since incomplete equilibration of water content after injecting aqueous tracer solution could lead to increased wetness near the injection spots and thus to enhanced denitrification (Wu et al., 2012). Hence, for the $^{15}$N gas-flux method a compromise must be found between homogeneous $^{15}$N label distribution, which is crucial for $N_2$ fluxes calculations, and a possibly minimal change of the real soil N transformations.

*Materials and Methods: L41: no rationale has been provided why the soil was sieved at 4 mm, and not e.g. at 2 mm, as commonly done.*

This is basically for simplification and fastening of sieving procedure. Silt loam soil is not easy to sieve and from our experience this only possible way to sieve large amounts of soil sufficient for experiments with large mesocosms as in our study with reasonable effort. We will expand the description to explain this:

4mm mesh size was used because this enabled us to sieve the necessary amount of soil (56 kg) within adequate time.

*L61: The ratio 30R should be 30N2/28N2, not 30N2/29N2*

Thank you this mistake will be corrected.

*L116: Not clear which differences in what were observed here.*

Differences in soil parameters presented in Table 2. This will be clarified

Significant differences in soil parameters between treatments (Table 2) were observed.

*L 136: "modulus of differences": Isn't the modulus the rest of a division?*

We meant by modulus the absolute value (not negative). We think this is right term, should we change to absolute value?

*L137: "Here it clear: : :": Unclear at this point, what is clear why.*

This will be clarified

For the comparison of mean absolute difference between $a_{P\_N2}$ and $a_{P\_N2O}$ we obtained quite a good agreement,

*L 138-139: ": : :much better than for comparisons with aNO3 (Table 3). This shows that both gases originate mostly from the same soil pool.": But the pool they originate from is the nitrate pool, isn't it? Shouldn't all three parameter be then comparable with each other?*

Yes, they should if the nitrate pool is homogenous. However, this is often not the case since we may deal with formation of isolated nitrate pools in soil especially in soil anoxic microsites. It was tested if one of the applied treatments may enhance this process. By this comparison it was shown that the bulk nitrite is not always representative for the pool where denitrification occurs.

*L146: ": : :than the aNO3 value measured for total soil.": The logic of this part of the sentence is not clear.*

This sentence will be clarified:

This shows that the multiple injection technique reduced the formation of isolated soil microsites characterized by distinct $^{15}N$ enrichment when compared to the bulk $a_{NO3}$ value measured.

*L160-162: Check wording, this sentence is hard to understand.*

This sentence will be clarified:

However, we can conclude that despite pronounced differences in $a^{15}N$ values of different treatments and different pools, the calculated results for gas fluxes and product ratios were mostly not significantly different between the treatments.

*L173-175: I would have expected the opposite logic here, i.e. that oxic conditions lead to greater disagreement due PRESENCE of nitrification and hence MORE dilution of the 15N-nitrate pool by native (soil-derived) N-sources.*

Yes, this is true. In the sentence we wrote about anoxic microsites that's why it was opposite. The sentence will be corrected to be easier to understand:

Oxic conditions can be expected to yield greater disagreement between $a_{NO3}$ and $a_P$ due to dilution of the bulk $a_{NO3}$ by soil-derived non-labelled N sources in contrast to anoxic soil microsites.

*L191-193: I think also here the logic is wrong. As it stands, the dominance of N2 fluxes is due to the calculation method applied.*

This sentence will be clarified:

This good accordance of the results is thanks to calculation method applying $a_P$ values determined individually for each sample which assures the adequate results for flux calculation, even with existence of multiple N pools.

*Figures general: I would not recommend the use of spline functions to connect the data points, but the use of straight lines instead.*

This will be modified.

*Fig. 1: Caption and figure panels do not fit together. Caption 1B says "fraction of 15Npool derived N2O", but Fig. 1B shows fp_N2, but the values are in ppm, which does not make sense (should be dimensionless between 0 and 1). Caption 1C says "N2 concentration", but Fig. 1C shows fp_N2+N2O, and again the values are in ppm, but should be dimensionless between 0 and 1.*

The inconsistency will be corrected: the figure caption will be modified. The ppm is correct: this is between 0 and 1, but it is a very low fraction expressed therefore in part per million. Fraction of labeled N2 is very low in atmospheric background, even since we used the modified atmosphere with only 2% of N2.

Figure 1: Comparison of the temporal changes in $N_2O$ concentration (A), fraction of $^{15}N$-pool derived $N_2$ (B), fraction of $^{15}N$-pool derived denitrification products ($N_2+N_2O$) (C), and $N_2O$ residual fraction (D) in three treatments: homogenized soil mixed with fertilizer (black dots), intact soil cores with fertilizer added through needle injection (red triangles), and homogenized soil with fertilizer added through needle injection (green squares). Error bars represent the standard deviation of 4 replicates within one treatment.

Thank you very much for the detailed edition of the manuscript in the attached supplement. All the corrections and suggestions will be taken into consideration by preparing the revised version of the manuscript.